# Association of total cholesterol variability with risk of venous thromboembolism: A nationwide cohort study

**Hyungjong Park[1]ⓔ, Yoonkyung Chang[2]ⓔ, Heajung Lee[3], Iksun Hong[3], Tae-Jin Song[3]\***

**1** Department of Neurology, Keimyung University, School of Medicine, Deagu, Republic of Korea,
**2** Department of Neurology, Mokdong Hospital, Ewha Womans University College of Medicine, Seoul, Republic of Korea, **3** Department of Neurology, Seoul Hospital, Ewha Womans University College of Medicine, Seoul, Republic of Korea

ⓔ These authors contributed equally to this work.

\* knstar@ewha.ac.kr

## Abstract

### Background

The effects of total cholesterol (TC) on coagulation and hemostatic systems could contribute to the development of venous thromboembolism (VTE). We investigated this possible association using TC variability.

### Methods

From the Korean NHIS-HEALS database, 1,236,589 participants with health screenings between 2003 and 2008 were included. TC variability was assessed using various parameters, including the coefficient of variation (CV), standard deviation (SD), and variability independent of mean (VIM). Occurrence of VTE was established by identifying at least two medical claims with a diagnostic code including various types of VTE: deep vein thrombosis (DVT) (I80.2–80.3), pulmonary embolism (PE) (I26, I26.0, I26.9), intraabdominal VTE (I81, I82, I82.2–82.3), and other VTE (I82.8–82.9).

### Results

Throughout the study's median follow-up period of 12.4 years (interquartile range 12.2–12.6) years, TC levels were assessed a total of 5,702,800 times. VTE occurred in 11,769 (1.08%) patients (DVT (4,708 (0.43%)), PE (3,109 (0.29%)), intraabdominal VTE (5,215 (0.48%)), and other VTE (4,794, (0.44%))). As a result, there was gradual association was observed between higher TC variability and occurrence of VTE. Multivariable analysis showed that quartile of TC variability using CV showed a positive correlation with the occurrence of VTE (adjusted hazard ratio (the highest versus lowest quartile), 1.14, 95% confidence interval, 1.08–1.20, p < 0.001). This result remained consistent applying to SD and VIM. In addition, higher quartile of TC variability was consistently associated with the development of various types of VTE in subgroup analysis.

**Data Availability Statement:** The NHIS-HEALS database was used in this study, but the data are not publicly available Requests for access can be submitted through the National Health Insurance

Sharing Service homepage [http://nhiss.nhis.or.kr/bd/ab/bdaba021eng.do]. To gain access, a completed application form, research proposal, and the approval from the IRB must be submitted to NHIS research support committee for review.

**Funding:** This project was supported by a grant from the Basic Science Research Program through the National Research Foundation of Korea funded by the Ministry of Education (2021R1F1A1048113 to T-JS). This work was supported by the Institute of Information & Communications Technology Planning & Evaluation (IITP) grant funded by the Korean government (MSIT) (2022-0-00621 to TJS, Development of artificial intelligence technology that provides dialog-based multi-modal explainability). This research supported by a grant from the Korea Health Technology R&D Project through the Korea Health Industry Development Institute (KHIDI), funded by the Ministry of Health & Welfare, Republic of Korea (grant number: HI22C073600, RS-2023-00262087 to TJS). The funding source had no role in the design, conduct, or reporting of this study.

**Competing interests:** The authors have declared that no competing interests exist.

## Conclusions

Increased TC variability may be associated with increased VTE risk. This analysis highlights the importance of maintaining stable TC levels to prevent the development of VTE.

## Introduction

B-type natriuretic peptide, troponin, and C-reactive protein and serum lipid profiles have known to be important biomarkers for cardiovascular (CV) event [1, 2]. The level of biomarkers in most biomarker studies has been determined through spot measurements, mainly because of cost constraints and the timing of blood sampling. However, this method was accompanied by reliability and reproducibility problems [1, 3, 4]. Therefore, recently, the growing interest in the variability of biomarkers within individual has highlighted their significant as risk factors for CV event [5].

VTE is a frequently occuring medical condition that imposes a substantial disease burden on a global scale and may lead to fatal outcomes in some cases [6]. The global shift towards an aging society is leading to a steady increased in the incidence of VTE [7]. Cancer, antiphospholipid syndrome (APS), fracture, obesity, renal failure and chronic inflammatory diseases are known risk factors for the development of VTE [7].

Furthermore, previous study showed a positive correlation between increased TC and DVT [8]. In addition, other studies have found a relationship between increased TC variability and end-stage renal disease and atrial fibrillation (AF) that are known to have relationship with the development of VTE [9, 10]. Therefore, there is a posibbility that TC variability may also contribute to the development of VTE. However, to date, there has been a scarcity of information on the association of TC variability and the occurrence of VTE. Our study was designed to explore the potential association between increased TC variability and an increased risk of VTE using nationwide population-based longitudinal cohort database.

## Materials and methods

### The information about data source

This study was based on the Korean National Health Insurance Service-National Health Screening (NHIS-HEALS) cohort database. The detailed information of NHIS-HEALS data was described in S1 Method.

### Study population

In the NHIS-HEALS database, participants who attended three or more health screenings that included TC level measurement between 2003 and 2008 were included (n = 1,236,589). The study excluded 91,231 participants due to the absence of data any variable necessary for this analysis. Participants (n = 4,414) with a previous VTE from January 2002 to the first health examination were also excluded. In addition, participants with less than three repeated measurements of TC level between 2003 and 2008 were excluded (n = 55,920). Finally, data from the remaining 1,085,024 participants were included. Fig 1 showed the selection process of participants.

### Definition of variables and comorbidities

The index date was the date of health examination. The following baseline characteristics were collected at the index date: age, sex, household income, and body mass index. Information on

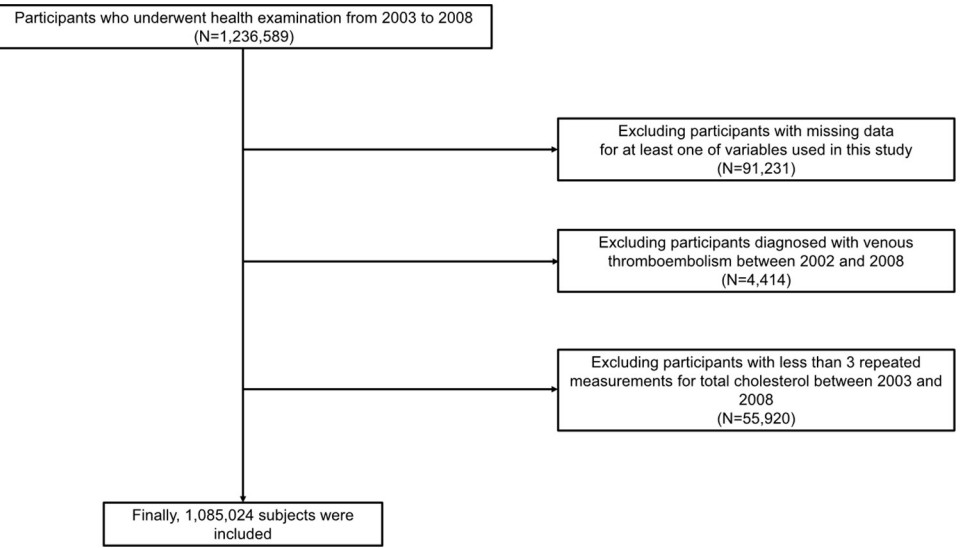

**Fig 1. Flow chart showing the participants selection process.**

smoking habits, alcohol consumption (frequency per week), and regular physical exercise (frequency per week) was obtained using questionnaires. Smoking status was categorized as none, former, and current smoker. Comorbidities were identified when present between January 2002 and the index date. The detailed definition of various comorbidities in our study was described in S1 Table [11–17].

## Definition of TC variability

TC variability was determined by assessing the intra-individual variability of the TC value at each examination performed during the six years before the index year (2009). Variability indices used in this study were coefficient of variation (CV), standard deviation (SD), and variability independent of the mean (VIM). The level of VIM was computed as $100 \times SD/Mean^{\beta}$, in which beta was the regression coefficient obtained from the natural logarithm of the ratio between SD and the mean [18].

## Study outcomes

The main outcome was the occurrence of VTE. Occurrence of VTE was established by identifying at least two claims with a diagnostic code including various types of VTE: DVT, (I80.2–80.3), pulmonary embolism (PE) (I26, I26.0, I26.9), intraabdominal VTE (I81-82, I82.2–82.3), and other VTE (I82.8–82.9)) and concurrent medication codes (anticoagulants and antiplatelet) based on a previous study [19]. Starting from index date, the follow-up period in this study extendeduntil the occurence of VTE, death of participants, or until the end of December 2020, whichever event occurred first.

## Statistical analysis

To compare the baseline characteristics among groups, categorical variables were analyzed using the Chi-square test, and continuous variable were analyzed using analysis of variance. The presentation of continuous variable was expressed as mean ±SD, while categorical variables were presented as the number and their percentages.

Restricted cubic splines were assessed to confirm the possibility of a non-linear association between TC variability and VTE risk, and all TC variabilities were fit for a positive linear association [20]. The participants were categorized into four groups according to TC variability quartiles with CV. Kaplan-Meier (KM) estimates with the log-rank test were used to evaluate the association of TC quartiles with TC for incident VTE risk. To estimate the incidence of VTE, the number of VTE cases was divided by the sum of person-years.

To determine the risk of quartiles of TC for VTE occurrence, Cox's proportional hazard regression was used; and hazard ratio (HR) and 95% confidence interval (CI) were determined. Multivariable regression models were constructed with (1) adjustment for age, sex, body mass index, household income, alcohol consumption, smoking status, regular physical activity, and presence of comorbidities (hypertension (HET), diabetes mellitus (DM), dyslipidemia, stroke, AF, cancer, renal disease, APS, and osteoporotic fracture and on lipid-lowering agent) and (2) adjusting variables in multivariable regression model (1) with mean TC level. The assumption of the proportionality of hazards was tested using Schoenfeld residuals. No departure from the proportional hazards' assumption was detected. For subgroup analysis, further analyses were performed for each type of VTE (DVT, PE, intraabdominal VTE, and other VTE).

For sensitivity analysis, CV, SD, and VIM were measured according to decile instead of quartile; lipid-lowering agents and mean TC level were further adjusted for in multivariable analysis; and participants with VTE within 1 year from the index date were excluded to minimize the possibility of reverse causality. The statistical analysis was performed using the Statistical Analysis System software (SAS version 9.2, SAS Institute, Cary, NC, USA). Results were considered statistically significant when the p-values were less than 0.05.

## Results

Table 1 demonstrates the results of the comparative analysis on the study population according to the quartiles of TC variability (CV). Participants with higher quartiles of TC variability were more commonly men and were older. These participants had higher frequencies of HET, DM, dyslipidemia, stroke, AF, cancer, renal disease, APS, and osteoporotic fracture.

During a median of 12.4 (interquartile range 12.2–12.6) years, VTE occurred in 11,769 (1.08%) patients. When each VTE was considered, DVT (4,708 (0.43%)), PE (3,109 (0.29%)), intraabdominal VTE (5,215 (0.48%)), and other VTE (4,794 (0.44%)) occurred. TC levels were measured a total of 5,702,800 times in this study. The number of participants undergoing measurement three times was 114,814; four times, 133,934; five times, 195,034; and six times, 641,242.

Fig 2A shows KM curves depicting the occurrence of VTE according to TC variability. The risk of incident VTE was dependent on quartiles of TC variability ($p < 0.001$). In multivariable analysis (1) adjusting for age, sex, body mass index, household income, alcohol consumption, smoking status, regular physical activity, and presence of comorbidities (HET, DM, dyslipidemia, stroke, AF, cancer, renal disease, AF, and osteoporotic fracture and on lipid-lowering agent), quartiles of TC variability assessed with CV were positively correlated with the occurrence of VTE (adjusted HR (the highest quartile versus lowest quartile), 1.14, 95% CI 1.08–1.20, $p < 0.001$, p for trend $< 0.001$). In addition, in multivariable model (2) adjusting variables in model (1) with mean TC level, quartiles of TC variables with CV were significantly associated with the occurrence of VTE (adjusted HR, 1.14, 95% CI 1.08–1.20, $p < 0.001$, p for trend $< 0.001$). This trend was consistent even when TC variability was applied to SD in model (1) (adjusted HR (the highest quartile vs lowest quartile), 1.13, 95% CI 1.08–1.20, $p < 0.001$, p for trend $< 0.001$) and model (2) (adjusted HR (the highest quartile versus lowest

**Table 1. Baseline characteristics of subjects according to total cholesterol variability.**

| Variable | Total | Q1 | Q2 | Q3 | Q4 | p-value |
|---|---|---|---|---|---|---|
| Number of participants (%) | 1085024 | 271256 (25.0) | 271256 (25.0) | 271256 (25.0) | 271256 (25.0) | |
| Age, years | 43.81±10.14 | 43.61±9.92 | 43.26±9.64 | 43.44±9.89 | 44.93±10.97 | < .001 |
| Sex | | | | | | < .001 |
| Male | 835845 (77.0) | 214649 (79.1) | 215495 (79.4) | 210546 (77.6) | 195155 (71.9) | |
| Female | 249179 (23.0) | 56607 (20.9) | 55761 (20.6) | 60710 (22.4) | 76101 (28.1) | |
| Body mass index (kg/m2) | 23.77±3.01 | 23.73±3.03 | 23.73±2.99 | 23.74±2.99 | 23.86±3.05 | < .001 |
| Household income | | | | | | < .001 |
| Q1, lowest | 158297 (14.6) | 36021 (13.3) | 34279 (12.6) | 38176 (14.1) | 49821 (18.4) | |
| Q2 | 338596 (31.2) | 79643 (29.4) | 81109 (29.9) | 85830 (31.6) | 92014 (33.9) | |
| Q3 | 396052 (36.5) | 101401 (37.4) | 103894 (38.3) | 100471 (37.0) | 90286 (33.3) | |
| Q4, highest | 192079 (17.7) | 54191 (20.0) | 51974 (19.2) | 46779 (17.3) | 39135 (14.4) | |
| Smoking status | | | | | | < .001 |
| Never | 557345 (51.4) | 136130 (50.2) | 134489 (49.6) | 137689 (50.8) | 149037 (54.9) | |
| Former | 161039 (14.8) | 41553 (15.3) | 41875 (15.4) | 40329 (14.9) | 37282 (13.7) | |
| Current | 366640 (33.8) | 93573 (34.5) | 94892 (35.0) | 93238 (34.4) | 84937 (31.3) | |
| Alcohol consumption (days/week) | | | | | | < .001 |
| None | 675686 (62.3) | 167922 (61.9) | 166655 (61.4) | 167706 (61.8) | 173403 (63.9) | |
| 1–4 | 390176 (36.0) | 99024 (36.5) | 100282 (37.0) | 98913 (36.5) | 91957 (33.9) | |
| ≥ 5 | 19162 (1.8) | 4310 (1.6) | 4319 (1.6) | 4637 (1.7) | 5896 (2.2) | |
| Regular physical activity (days/week) | | | | | | < .001 |
| None | 467480 (43.1) | 114779 (42.3) | 114050 (42.1) | 115990 (42.8) | 122661 (45.2) | |
| | 549408 (50.6) | 139688 (51.5) | 140611 (51.8) | 138401 (51.0) | 130708 (48.2) | |
| ≥ 5 | 68136 (6.3) | 16789 (6.2) | 16595 (6.1) | 16865 (6.2) | 17887 (6.6) | |
| Comorbidities | | | | | | |
| Hypertension | 213936 (19.7) | 46490 (17.1) | 46378 (17.1) | 50214 (18.5) | 70854 (26.1) | < .001 |
| Diabetes mellitus | 107940 (10.0) | 21533 (7.9) | 21808 (8.0) | 24466 (9.0) | 40133 (14.8) | < .001 |
| Dyslipidemia | 210776 (19.4) | 37817 (13.9) | 41250 (15.2) | 48543 (17.9) | 83166 (30.7) | < .001 |
| Stroke | 10752 (1.0) | 1949 (0.7) | 1941 (0.7) | 2287 (0.8) | 4575 (1.7) | < .001 |
| Atrial fibrillation | 4134 (0.4) | 794 (0.3) | 795 (0.3) | 881 (0.3) | 1664 (0.6) | < .001 |
| Renal disease | 12788 (1.2) | 2165 (0.8) | 2326 (0.9) | 2694 (1.0) | 5603 (2.1) | < .001 |
| Cancer | 23953 (2.2) | 5132 (1.9) | 5166 (1.9) | 5748 (2.1) | 7907 (2.9) | < .001 |
| Antiphospholipid syndrome | 3981 (0.4) | 811 (0.3) | 808 (0.3) | 939 (0.4) | 1423 (0.5) | < .001 |
| Osteoporotic fracture | 15341 (1.4) | 3589 (1.3) | 3355 (1.2) | 3621 (1.3) | 4776 (1.8) | < .001 |
| On lipid-lowering agent | 99065 (9.1) | 17774 (6.6) | 21038 (7.8) | 25242 (9.3) | 44078 (16.2) | < .001 |
| Mean total cholesterol (mg/dL) | 191.93±30.88 | 192.30±29.43 | 191.38±29.17 | 190.90±29.29 | 193.14±35.16 | < .001 |
| Total cholesterol variability | | | | | | |
| CV (%) | 9.54±5.46 | 4.77±1.22 | 7.55±0.65 | 10.00±0.81 | 15.83±7.08 | < .001 |
| SD | 18.53±19.83 | 9.17±2.75 | 14.44±2.53 | 19.09±3.32 | 31.43±35.73 | < .001 |
| VIM (%) | 18.56±19.74 | 9.16±2.74 | 14.41±2.51 | 19.11±3.43 | 31.45±35.77 | < .001 |

*p*-value by Chi-square test. Data are expressed as the mean ± SD, or n (%).

Q, Quartile, CV, coefficient of variation; SD, standard deviation; VIM, variability independent of the mean.

quartile), 1.15, 95% CI 1.09–1.21, P < 0.001, p for trend < 0.001) (Table 2). In addition, TC variability was also consistently associated with VIM in model (1) (adjusted HR (the highest quartile versus lowest quartile), 1.13, 95% CI, 1.08–1.20, p < 0.001, p for trend < 0.001) and model (2) (adjusted HR, 1.15, 95% CI, 1.09–1.21, p < 0.001, p for trend < 0.001) (**Table 2**). These findings were consistent regardless of sex (**S2 and S3** Tables).

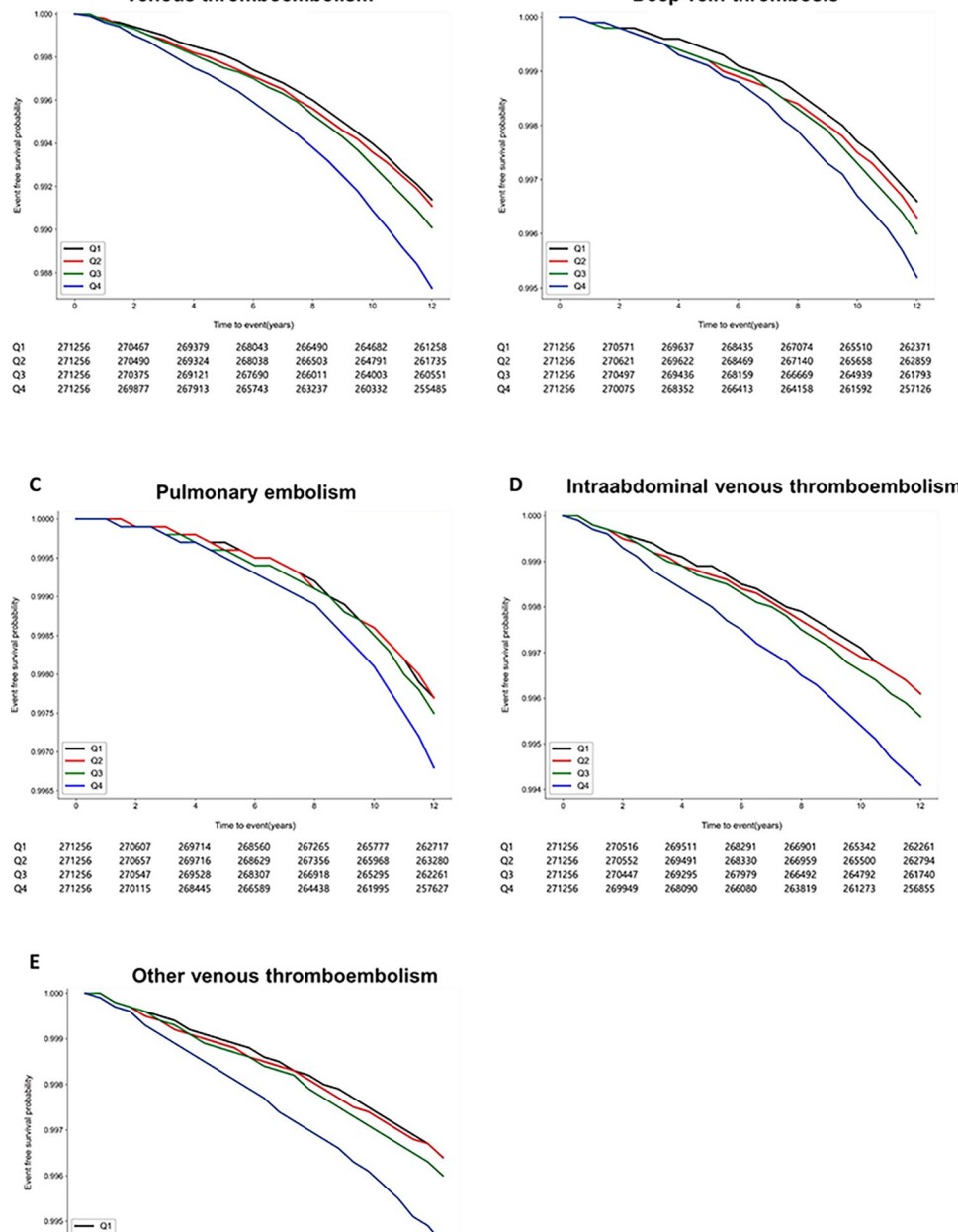

**Fig 2. Kaplan-Meier survival curves for occurrence of venous thromboembolism (VTE) according to TC variability.** (A) All types of VTE. (B) Deep vein thrombosis, (C) Pulmonary embolism, (D) Intraabdominal VTE, (E) Other VTE.

For sensitivity analysis, the association between quartiles of TC variability and the occurrence of VTE was consistently noted even when the use of lipid-lowering agents and mean TC levels were additively adjusted (S4 Table). In addition, association of deciles of TC variability and the development of VTE remained consistent (S5 Table).

**Table 2. The risk for occurrence of venous thromboembolism according to quartiles of total cholesterol variability.**

| | Number of participants | Number of events | Event rate (%) (95% CI) | Person-years | Incidence rate (per 1000 person-years) | Multivariable model (1) Adjusted HR (95% CI) | p-value | p-value for trend | Multivariable model (2) Adjusted HR (95% CI) | p-value | p-value for trend |
|---|---|---|---|---|---|---|---|---|---|---|---|
| CV | | | | | | | | < .001 | | | < .001 |
| Q1 | 271256 | 2546 | 0.94 (0.90, 0.98) | 3315661.54 | 0.77 | 1 (reference) | | | 1 (reference) | | |
| Q2 | 271256 | 2587 | 0.95 (0.92, 0.99) | 3317715.89 | 0.78 | 1.05 (0.99, 1.11) | 0.086 | | 1.05 (0.99, 1.11) | 0.092 | |
| Q3 | 271256 | 2892 | 1.07 (1.03, 1.11) | 3312197.41 | 0.87 | 1.12 (1.06, 1.18) | < .001 | | 1.12 (1.06, 1.18) | < .001 | |
| Q4 | 271256 | 3743 | 1.38 (1.34, 1.42) | 3283556.48 | 1.14 | 1.14 (1.08, 1.20) | < .001 | | 1.14 (1.08, 1.20) | < .001 | |
| SD | | | | | | | | < .001 | | | < .001 |
| Q1 | 271272 | 2376 | 0.88 (0.84, 0.91) | 3317874.78 | 0.72 | 1 (reference) | | | 1 (reference) | | |
| Q2 | 271219 | 2575 | 0.95 (0.91, 0.99) | 3316510.75 | 0.78 | 1.07 (1.01, 1.13) | 0.021 | | 1.07 (1.01, 1.13) | 0.014 | |
| Q3 | 271288 | 2909 | 1.07 (1.03, 1.11) | 3310978.22 | 0.88 | 1.11 (1.05, 1.17) | < .001 | | 1.12 (1.06, 1.18) | < .001 | |
| Q4 | 271245 | 3908 | 1.44 (1.40, 1.49) | 3283767.58 | 1.19 | 1.13 (1.08, 1.20) | < .001 | | 1.15 (1.09, 1.21) | < .001 | |
| VIM | | | | | | | | < .001 | | | < .001 |
| Q1 | 271256 | 2376 | 0.88 (0.84, 0.91) | 3317677.94 | 0.72 | 1 (reference) | | | 1 (reference) | | |
| Q2 | 271256 | 2575 | 0.95 (0.91, 0.99) | 3316969.08 | 0.78 | 1.07 (1.01, 1.13) | 0.022 | | 1.07 (1.01, 1.13) | 0.014 | |
| Q3 | 271256 | 2909 | 1.07 (1.03, 1.11) | 3310591.10 | 0.88 | 1.11 (1.05, 1.17) | < .001 | | 1.12 (1.06, 1.18) | < .001 | |
| Q4 | 271256 | 3908 | 1.44 (1.40, 1.48) | 3283893.20 | 1.19 | 1.13 (1.08, 1.21) | < .001 | | 1.15 (1.09, 1.22) | < .001 | |

Multivariable model (1) was adjusted for sex, age, body mass index, household income levels, smoking, alcohol consumption, regular physical activity, hypertension, diabetes mellitus, dyslipidemia, stroke, atrial fibrillation, renal disease, cancer, antiphospholipid syndrome, osteoporotic fracture and on lipid-lowering agent.

Multivariable model (2) was adjusted for sex, age, body mass index, household income levels, smoking, alcohol consumption, regular physical activity, hypertension, diabetes mellitus, dyslipidemia, stroke, atrial fibrillation, renal disease, cancer, antiphospholipid syndrome, osteoporotic fracture, on lipid-lowering agent and mean total cholesterol.

CI, confidence interval; HR, hazard ratio, CV, coefficient of variation; Q, Quartile; SD, standard deviation; VIM, variability independent of the mean

Moreover, even when the occurrence of VTE was redefined as 1 year after the index date using landmark analysis, the association of TC variability with the occurrence of VTE continued to be consistent when using both quartiles (**S6 Table**) and deciles of TC variability (**S7 Table**). In subgroup analysis, the highest quartiles of TC variability were significantly and positively associated with the risk of DVT (**S8 Table**, **Fig 2B**), PE (**S9 Table**, **Fig 2C**), intraabdominal VTE (**S10 Table**, **Fig 2D**), and other VTE (**S11 Table**, **Fig 2E**) compared to the lowest quartiles of TC variability regardless of parameters for variability (CV, SD, and VIM), even after adjusting for lipid-lowering agents and mean TC levels.

## Discussion

The key findings of our study were that TC variability increased the risk of VTE occurrence; and the impact of TC variability was consistent regardless of the different parameters of variability and VTE locations, including DVT, PE, intraabdominal VTE, and other VTE.

Recently, variability in biological parameters has been emphasized as a risk factor CV morbidity and mortality that was not previously acknowledged [5]. This finding was based on the hypothesis that loss of physiological homeostasis through disease would lead to disturbances in intrinsic variability [21]. In this regard, a post-hoc analysis from the Treating to New Target (TNT) trial demonstrated that variability in low density lipoprotein cholesterol (LDL-C) level between visits were a predictor of any coronary event, CV event, myocardial infarction (MI), death, and stroke [22]. In another study of patients with ST-segment elevation MI, both increased variability of LDL-C and high-density lipoprotein cholesterol (HDL-C) indices were linked to the increased development of major adverse CV events [23]. In addition, TC variability was associated with coronary atheroma progression, poor clinical outcome, and the development of chronic kidney disease and AF [9, 10, 24]. Accordingly, not only increased levels of TC, LDL-C, and low LDL-C using spot measurement, but also variability of lipid profile has been regarded as established additional risk factors for the various CV diseases.

VTE is a significant global health issue, with several risk factors associated with its development, including malignancy, older age, immobilization, obesity, recent fracture, surgery and hematologic heritable disorders [7]. In addition, both idiopathic and secondary VTE are known to have coexisting risk factors for CV diseases such as HET, DM, smoking and dyslipidemia [25].

Dyslipidemia, in particular, has been postulated to contribute to the development of VTE due to the influence of lipids and lipoproteins on hemostasis by modulating the expression and function of procoagulants, fibrinolytic, and theological factors [26]. However, studies investigating the association between lipid profiles and the development of VTE have yielded inconsistent results, with most studies relying only on spot measurement of lipid levels [27].

However, our study utilized TC variability, which is a more reliable measure than spot measurements. We found that TC variability had a positive impact on the development of VTE, regardless of different parameters for TC variability, including CV, SD and VIM. Furthermore, in subgroup analyses, TC variability consistently showed a positive impact on the development of various type of VTE, including DVT and PE, which have high mortality and recurrence rates [28]. Our finding suggested that assessing TC variability may be important for patients at high risk of VTE, such as those with malignancy, obesity, immobilization, and the use of oral contraceptives [7]. This caution could help to prevent fatal PE or DVT.

One theoretical explanation for the association between TC variability, indicating an abnormal and unstable TC level, and VT could be the regulation of coagulation impairment, including the thrombomodulin (TM) and tissue factor pathway inhibitor (TFPI) [29]. TFPI, is an anticoagulant protein that inhibits the procoagulant pathway [30]. The unbound form of TFPI is transferred to the LDL-C in a hypercholesterolemia state, and this transfer consequently causes a reduction in endothelial cell-associated TFPI and leads to a hypercoagulable status [30]. In addition, the activity of TM, which has an anticoagulant effect after binding to thrombin, was decreased in the associated study [31]. Thereafter, plasma levels of TM as free form, the marker of damaged endothelial cells, were significantly increased in the hypercholesterolemia state [29]. Thus, TC variability may be implicated in the pathogenesis of VTE by interrupting the coagulation cascade and causing endothelial dysfunction. Finally, abnormal TC level induces pro-inflammatory conditions [32]. Inflammation, an established risk factor for VTE [25], contributes to the development of endothelial dysfunction and has been linked to the development of VTE.

This study had several limitations. First, there is the possibility of residual that may be associated with the development of VTE, such as use of oral contraceptive, the presence of inherited hematologic disorders, or D-dimer levels [7]. However, data on these variables could not be obtained from the current dataset. Furthermore, the presence of dyslipidemia showed a

similar positive impact on the development of idiopathic and secondary DVT in previous study [25]. Second, the study subjects only included Koreans, and the results may be different in other races/ethnicities. External validation of our results in other ethnicities is needed. Third, despite sufficient evidence of TC variability in CV diseases, a clear cause of TC variability has not been fully elucidated. Fourth, considering the observational and retrospective nature of our study, the association between TC variability and VTE could not be causal. However, to minimize the possibility of a reverse causal relationship, participants with previous history of VTE were excluded. In addition, a sensitivity analysis that excluded participants with VTE within one year from the index date using landmark analysis showed similar results. Fifth, patients with cerebral venous thromboembolism (CVT) were not included. CVT is more prevalent in young-aged women [33], and our study included patients aged ≥ 40 years. Thus, the influence of excluding CVT was probably quite low. Sixth, we calculated TC variability using TC levels from annual health screening. Thus, real time variability of TC could not be reflected. Finally, the mean age of target population was skewed toward lower age margin of target population. This could be originated from the lower participation rates for health screening program in elderly [34].

This study had several strengths. This is the first study on the impact of TC variability on the development of VTE using large-scale, nationally representative data that have been tracked over a long time. Our results suggest unique evidence supporting the benefits of maintaining TC levels for VTE prevention and the need to focus on TC variability in addition to other known risk factors of VTE. Despite the importance of TC variability on CV diseases and the development of VTE, clinical management to achieve stable TC variability has not been revealed. Therefore, further studies are necessary.

## Conclusion

In this nationwide population-based cohort study, we observed that greater TC variability was significantly associated with the development of VTE. This relationship was consistently observed regardless of types of VTE. Future studies should examine whether reducing TC variability can decrease the development of VTE.

## Supporting information

**S1 Method. Information about data source.**
(DOCX)

**S1 Table. Definition of covariates.**
(DOCX)

**S2 Table. The risk of the occurrence of venous thromboembolism according to the quartile of total cholesterol variability in men.**
(DOCX)

**S3 Table. The risk of the occurrence of venous thromboembolism according to the quartiles of total cholesterol variability.**
(DOCX)

**S4 Table. The risk of venous thromboembolism according to quartiles of total cholesterol variability additionally adjusting the use of lipid lowering agents and mean total cholesterol level.**
(DOCX)

**S5 Table. The risk of occurrence of venous thromboembolism according to deciles of total cholesterol variability.**
(DOCX)

**S6 Table. The risk of occurrence of venous thromboembolism according to quartiles of total cholesterol variability (landmark analysis).**
(DOCX)

**S7 Table. The risk of occurrence of venous thromboembolism according to deciles of total cholesterol variability (landmark analysis).**
(DOCX)

**S8 Table. The risk of occurrence of deep vein thrombosis according to quartiles of total cholesterol variability.**
(DOCX)

**S9 Table. The risk of occurrence of pulmonary embolism according to quartiles of total cholesterol variability.**
(DOCX)

**S10 Table. The risk of occurrence of intraabdominal venous thromboembolism according to quartiles of total cholesterol variability.**
(DOCX)

**S11 Table. The risk of occurrence of other venous thromboembolism according to quartiles of total cholesterol variability.**
(DOCX)

## Author Contributions

**Conceptualization:** Hyungjong Park, Yoonkyung Chang, Tae-Jin Song.

**Data curation:** Yoonkyung Chang, Tae-Jin Song.

**Formal analysis:** Heajung Lee, Iksun Hong.

**Investigation:** Heajung Lee, Iksun Hong.

**Writing – original draft:** Hyungjong Park, Yoonkyung Chang, Tae-Jin Song.

**Writing – review & editing:** Tae-Jin Song.

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
