## [Decision Letter · Decision Letter 0]

27 Feb 2023

PONE-D-22-29922Association of total cholesterol variability with risk of venous thrombosis: a nationwide cohort studyPLOS ONE

Dear Dr. Song,

Thank you for submitting your manuscript to PLOS ONE. After careful consideration, we feel that it has merit but does not fully meet PLOS ONE’s publication criteria as it currently stands. Therefore, we invite you to submit a revised version of the manuscript that addresses the points raised during the review process and you will find below in the "comments to the authors" section.

We look forward to receiving your revised manuscript.

Kind regards,

Christophe Leroyer

Academic Editor

PLOS ONE

Journal Requirements:

NO authors have competing interests

**Comments to the Author**

1. Is the manuscript technically sound, and do the data support the conclusions?

Reviewer #1: Yes

2. Has the statistical analysis been performed appropriately and rigorously? 

Reviewer #1: Yes

3. Have the authors made all data underlying the findings in their manuscript fully available?

Reviewer #1: Yes

4. Is the manuscript presented in an intelligible fashion and written in standard English?

Reviewer #1: No

5. Review Comments to the Author

Reviewer #1: This manuscript reports on a population study on the association between variability of total cholesterol (TC) variability over time and the risk for venous thromboembolism (VTE). TC variability was measured using the coefficient of variation (CV), standard deviation (SD), and variability independent of mean (VIM). All these parameters were consistently and independently associated with a 15% increased risk for VTE.

This is an interesting manuscript. Prior to publication, I suggest the authors to address the following:

1) the age of the target population ranged from 40 to 79 years (line 90). How do the authors explain that the mean age of the studied population was only 43.8±10.1? The age distribution looks very close to the lower end of the targeted population and the results may only apply to a younger population as compared with older, which is at higher risk for both VTE and high TC.

2) have the authors considered separating analysis according to sex (as sex is an important risk factor for cardiovascular diseases)?

3) CV, SD, and VIM are collinear and based on the same parameters (mean, SD). It lead to the same risk estimates for each parameter. Could this be discussed?

4) the article needs editing throughout:

First, I suggest using “VTE” instead of “VT” for venous thromboembolism, as it is a more common abbreviation. Also, please use “DVT” for deep vein thrombosis and “PE” for pulmonary embolism.

The manuscript is sometimes unclear or imprecise, particularly the introduction (the first sentence is hard to understand, second paragraph on risk factors should be edited (major risk factors for VTE are surgery, immobilisation/fracture of lower limbs, cancer, APS; all other risk factors are at intermediate risk for VTE; I would tone done the association between AFib, renal insufficiency and VTE as well in paragraph 3).

Consider putting all criteria for definition of cardiovascular risk factors and diseases in an appendix instead of listing all ICD codes etc.

Line 274 to 296 should be rephrased. Line 290 to 293: it is not clear whether the authors refer to their papers or others.

6. PLOS authors have the option to publish the peer review history of their article (what does this mean?). If published, this will include your full peer review and any attached files.

Reviewer #1: No

---

## [Author Response · Author response to Decision Letter 0]

7 Apr 2023

We have written all the reviewer's comment in the attached file. Thank the reviwer's valuable comment, again.

---

## [Editor Report · Decision Letter 1]

26 Jul 2023

Association of total cholesterol variability with risk of venous thromboembolism: a nationwide cohort study

PONE-D-22-29922R1

Dear Dr. Song,

We’re pleased to inform you that your manuscript has been judged scientifically suitable for publication and will be formally accepted for publication once it meets all outstanding technical requirements.

Kind regards,

Christophe Leroyer

Academic Editor

PLOS ONE

---

## [Editor Report · Acceptance letter]

8 Aug 2023

PONE-D-22-29922R1 

Association of total cholesterol variability with risk of venous thromboembolism: a nationwide cohort study 

Dear Dr. Song:

I'm pleased to inform you that your manuscript has been deemed suitable for publication in PLOS ONE. Congratulations! Your manuscript is now with our production department. 

Kind regards, 

on behalf of

Dr. Christophe Leroyer 

Academic Editor

PLOS ONE